# Therapeutic Efficacy of GC1118, a Novel Anti-EGFR Antibody, against Glioblastoma with High EGFR Amplification in Patient-Derived Xenografts

**DOI:** 10.3390/cancers12113210

**Published:** 2020-10-31

**Authors:** Kyoungmin Lee, Harim Koo, Yejin Kim, Donggeon Kim, Eunju Son, Heekyoung Yang, Yangmi Lim, Minkyu Hur, Hye Won Lee, Seung Won Choi, Do-Hyun Nam

**Affiliations:** 1Department of Health Science and Technology, SAIHST, Sungkyunkwan University, Seoul 06355, Korea; kyoungmin@g.skku.edu (K.L.); guhalim@skku.edu (H.K.); yiejin89@g.skku.edu (Y.K.); 2Institute for Refractory Cancer Research, Samsung Medical Center, Seoul 06351, Korea; donggeon.kim@gmail.com (D.K.); sej8749@gmail.com (E.S.); heekyoungyang@gmail.com (H.Y.); 3MOGAM Institute for Biomedical Research, Yongin 16924, Korea; ymlin@mogam.re.kr (Y.L.); zymogam@mogam.re.kr (M.H.); 4Department of Hospital Medicine, Yonsei University College of Medicine, Seoul 03722, Korea; nsproper@yuhs.ac; 5Department of Neurosurgery, Sungkyunkwan University, School of Medicine, Samsung Medical Center, Seoul 06351, Korea

**Keywords:** glioblastoma, epidermal growth factor receptor, monoclonal, xenograft, amplification

## Abstract

**Simple Summary:**

GC1118 is a novel anti-EGFR monoclonal antibody with a distinct mode of epitope binding. Its therapeutic efficacy has been validated in preclinical studies of several cancers. We evaluated the anti-tumor efficacy of GC1118 against glioblastoma (GBM) using patient-derived xenografts (PDXs). GC1118 exhibited anti-tumor efficacy comparable to that of cetuximab in a subset of PDXs, and *EGFR* amplification was a potential biomarker for predicting its therapeutic efficacy. Growth inhibitory and direct apoptotic effects on GBM tumor cells were confirmed in in vitro analyses. In intracranial PDXs, GC1118 significantly improved survival outcome, indicating its potential to cross the blood–brain barrier. These results support the clinical potential of GC1118 in treating GBM, further prompting the requirement of a clinical trial.

**Abstract:**

We aimed to evaluate the preclinical efficacy of GC1118, a novel anti-epidermal growth factor receptor (EGFR) monoclonal antibody (mAb), against glioblastoma (GBM) tumors using patient-derived xenograft (PDX) models. A total of 15 distinct GBM PDX models were used to evaluate the therapeutic efficacy of GC1118. Genomic data derived from PDX models were analyzed to identify potential biomarkers associated with the anti-tumor efficacy of GC1118. A patient-derived cell-based high-throughput drug screening assay was performed to further validate the efficacy of GC1118. Compared to cetuximab, GC1118 exerted comparable growth inhibitory effects on the GBM tumors in the PDX models. We confirmed that GC1118 accumulated within the tumor by crossing the blood–brain barrier in in vivo specimens and observed the survival benefit in GC1118-treated intracranial models. Genomic analysis revealed high *EGFR* amplification as a potent biomarker for predicting the therapeutic efficacy of GC1118 in GBM tumors. In summary, GC1118 exerted a potent anti-tumor effect on GBM tumors in PDX models, and its therapeutic efficacy was especially pronounced in the tumors with high *EGFR* amplification. Our study supports the importance of patient stratification based on *EGFR* copy number variation in clinical trials for GBM. The superiority of GC1118 over other EGFR mAbs in GBM tumors should be assessed in future studies.

## 1. Introduction

The epidermal growth factor receptor (EGFR) is a major oncogenic driver in glioblastoma (GBM) [1]. Up to 60% of GBM tumors harbor *EGFR* alterations, including amplifications and mutations [1,2]. Given its frequency and specific expression confined to tumors, EGFR has been considered as a notable therapeutic target for GBM tumors.

EGFR tyrosine kinase inhibitors (TKIs) have shown successful clinical outcomes in various solid cancers harboring *EGFR* mutations, especially lung cancer. Treatment with a third generation EGFR TKI, Osimertinib, remarkably improved the outcomes of patients with refractory non-small cell lung cancer (NSCLC) harboring resistance mutations (T790M) [3]. Osimertinib also demonstrated considerable potential of penetrating the blood–brain barrier (BBB), indicating that it can possibly target central nervous system (CNS) metastases. However, despite the success in patients with lung cancer, GBM has not benefitted from the use of EGFR TKIs [4,5,6]. There are many possible explanations for treatment failure, and the distinct genomic landscape of *EGFR*-mutated GBM tumors is one of them.

Unlike in NSCLCs, sensitizing *EGFR* mutations for known EGFR TKIs are rarely found in GBMs. The EGFRvIII variant and *EGFR* amplifications are the most notable *EGFR* genetic alterations in GBM [2,7]. Dynamic regulation of extrachromosomal *EGFR* DNA content could also account for treatment resistance toEGFR TKIs in GBM [8]. Therefore, directly blocking the receptors rather than their intracellular TK domain might be a more effective therapeutic strategy against GBMs.

Several anti-EGFR monoclonal antibodies (mAbs) (e.g., cetuximab and, nimotuzumab, etc.) have been developed. However, none of them has considerably improved the prognosis of patients with GBM [9]. Cetuximab failed to demonstrate efficacy for recurrent GBM [10,11,12], and nimotuzumab did not confer a significant survival benefit in a phase III trial [13]. Nimotuzumab only provided clinical benefits to a minor subset of patients (newly diagnosed GBM with unmethylated pMGMT). GC1118 is a novel anti-EGFR mAb that binds to an epitope that is distinct from those of several commercially available EGFR mAbs. Compared to cetuximab, GC1118 has shown to exhibit enhanced binding affinity to both low-affinity and high-affinity EGFR ligands. Its ability to bind to high-affinity EGFR ligands (e.g., HB-EGF, TGFα, BTC, and EGF) contributes to its anti-tumor efficacy against *KRAS*-mutant colorectal cancer (CRC) in a patient-derived xenograft (PDX) model resistant to cetuximab treatment [14].

Lee et al. observed an elevation in the level of high-affinity EGFR ligands in *KRAS*-mutant CRC PDXs and GC1118 exerted better anti-tumor effects in these models than cetuximab, thereby highlighting its unique therapeutic implication [14]. In another study, GC1118 exhibited anti-tumor effects against gastric cancer irrespective of *KRAS* mutation [15]. The authors also observed the synergistic effect of a combinatorial approach involving GC1118 and a conventional cytotoxic agent. All of these results indicate the potential of GC1118 as an anti-tumor agent.

We hypothesized that GC1118 could be a potential therapeutic agent for GBMs owing to its ability to bind to a distinct epitope. We aimed to evaluate the preclinical efficacy of GC1118 for GBM tumors using PDXs. We also evaluated whether GC1118 could cross the blood–brain barrier (BBB) to exert anti-tumor effects in GBM tumors of intracranial xenograft models. Considering the enormous genomic heterogeneity of GBMs, genomic biomarkers associated with the therapeutic efficacy of GC1118 were also investigated.

## 2. Results

### 2.1. GC1118, a Novel Anti-EGFR Monoclonal Antibody, Exhibited Anti-Tumor Effects in Patient-Derived GBM Xenograft Models

PDX models were established to evaluate the anti-tumor efficacy of GC1118 in GBM tumors. Patient-derived GBM tumor cells were implanted in the unilateral flank of BALB/c nude mice. We adopted a single-mouse trial scheme for establishing PDX models to recapitulate the real clinical trial setting [16] (Figure 1A). Fifteen distinct patient-derived GBM tumor cells were used to establish PDX models, and three mice were implanted with cells from each patient.

GC1118 exerted its anti-tumor effects in eight PDXs (G022, G677, B802, G608, G542, G096, G698, and G500) but failed to do so in the remaining PDXs (G317, B849, B823, B891, G316, B930, and G775) (Figure 1B). To quantify the effects, the tumor inhibition rate (TIR) was defined as follows (1):TIR = [{1-(volume of treated tumor)/(volume of control tumor)} × 100(%)](1)

When the TIR exceeded 50%, the corresponding PDXs were categorized as sensitive to treatment (Figure 1C). None of the PDXs showed side effects specific for cetuximab or GC1118. To understand the molecular traits underlying these different anti-tumor effects, we evaluated the expression of wild-type EGFR and EGFRvIII, two notable alterations frequently detected in GBMs (Figure 1D). EGFR was overexpressed in the GC1118-sensitive group (EGFR expression was normalized to that of β-actin; 1.00 ± 0.28 vs. 0.73 ± 0.24, for the GC1118-sensitive vs. -resistant group, respectively, *p*-value = 0.69, two-sided Wilcoxon rank-sum test). EGFRvIII was expressed in three PDXs from the GC1118-sensitive group, but not in any PDXs from the GC1118-resistant group (*p*-value = 0.20, Fisher’s exact test).

### 2.2. Anti-Proliferative Effect of GC1118 on GBM Patient-Derived Cells (PDCs)

The anti-proliferative effect of GC1118 on GBM tumors was evaluated using the in vitro 3-(4,5-dimethylthiazol-2-yl)-2,5-diphenyl tetrazolium bromide (MTT) assay with PDCs. Three different PDCs were used (G096, G022, and G775). Cell growth was suppressed in G096 and G022 PDCs, but not in the G775 PDCs (Figure 2). These results were consistent with in vivo findings; the PDXs of G096 and G022 were sensitive to GC1118, while the PDX of G775 was resistant.

### 2.3. EGFR Amplification as a Potent Biomarker to Predict the Therapeutic Efficacy of GC1118 in GBM Tumors

In vivo experiments confirmed the anti-tumor efficacy of GC1118 against GBM tumors. However, the efficacy was selective for a subset of GBM PDXs. Therefore, we examined the mutational landscape of PDX models to identify genomic biomarkers for predicting GC1118 sensitivity.

Previous clinical trials involving EGFR-targeting agents have provided several important insights. Redundant signaling pathways in GBMs have contributed significantly to the failure of many clinical trials [17]. We assessed the mutational status of relevant genes in these redundant pathways (e.g., RAS/MAPK, ERK, and PTEN/PI3K); however, we failed to identify any predictors of GC1118 sensitivity (Figure 3A). The PTEN/PI3K signaling cascade is one of the important downstream pathways that might reverse the effects of enhanced EGFR signaling. *PTEN* deletion and *PIK3CA* mutations were observed at a similar frequency in both GC1118-sensitive and -resistant PDXs (*PTEN*, 75% (*n* = 6/8) vs. 66.7% (*n* = 4/6); *PIK3CA*, 12.5% (*n* = 1/8) vs. 16.7% (*n* = 1/6)). Unlike that in CRC, genes involved in RAS signaling were rarely mutated in GBMs. Only one GBM PDX of the GC1118-sensitive group harbored a missense mutation in *NRAS* gene. No PDXs harbored any alteration in *PDGFR* or *MET* (Figure 3A).

Interestingly, gain-of-function *EGFR* alterations, which include both amplification and a gain-of-function mutation, can potentially indicate the sensitivity of GBM PDXs to GC1118 (odds ratio (OR), 10.98 (95% CI, 0.64–779.16), *p*-value = 0.09, Fisher’s exact test). Five GBM PDXs in the GC1118-sensitive group harbored an *EGFR* amplification, and two PDXs without amplification harbored gain-of-function *EGFR* mutations (V774M and T790M (B802), C636W (G542)). There were only two PDXs with *EGFR* amplification or gain-of-function mutations in the resistant group. PDXs of the sensitive group exhibited higher *EGFR* amplification compared to those of the resistant group (log2 value of *EGFR* copy number, 2.74 ± 0.80 vs. 1.23 ± 0.93, for GC1118-sensitive vs. -resistant group, respectively, *p*-value = 0.543, two-sided Wilcoxon rank sum test) (Figure 3B).

Next, we investigated the anti-tumor activity of GC1118 using PDC-based high-throughput drug screening. PDCs seeded in 384-well plates were treated with seven different doses of GC1118, and cell viability was assessed as a measure of the area under the dose–response curve (AUC). The median value of the AUC was used to categorize PDCs into GC1118-sensitive and -resistant subsets. PDCs with an AUC above the median AUC value were sensitive to GC1118. Thirty-six different PDCs were screened, 16 of which were sensitive to GC1118. The *EGFR* copy number was significantly higher in GC1118-sensitive PDCs (log2 value of *EGFR* copy number, 2.9 ± 0.56 vs. 0.90 ± 0.30, for GC1118-sensitive vs. -resistant group, respectively, *p*-value = 0.003, two-sided Wilcoxon rank sum test), and *EGFR* amplification status was significantly associated with the response to GC1118 (OR = 6.21 (95% CI, 1.26–36.87), *p*-value = 0.017, Fisher’s exact test)) (Figure 3C).

### 2.4. GC1118 Crossed the Blood–Brain Barrier, Thereby Conferring Survival Benefit in Orthotopic PDX Models

BBB penetration is one of major concerns in mAb-based therapy for brain tumors [18]. The limited efficacy of cetuximab in patients with *EGFR*-amplified GBM was partially attributed to its limited BBB penetration [19].

We established an intracranial PDX model to investigate whether GC1118 can readily cross the BBB and improve survival outcome. PDXs treated with GC1118 showed significantly better survival than the control group (median overall survival, 150 days (GC1118-treated) vs. 70 days (control), *p*-value = 0.005, two-sided Log-rank test)) (Figure 4A). Although not statistically significant, GC1118-treated PDXs showed better survival than PDXs treated with cetuximab (median overall survival, 120 days for cetuximab-treated PDXs, *p*-value = 0.16, two-sided Log-rank test).

To detect the presence of GC1118 in the brain, mice were euthanized at several time points (62,448, and 72 h) after intraperitoneal injection, and the brain slides were stained with hematoxylin and eosin (H&E) and labeled with a goat anti-human secondary antibody for performing immunohistochemistry (IHC). We observed that a significant amount of GC1118 accumulated in the tumor core, suggesting that GC1118 can reach the tumor by crossing both BBB and brain-tumor barrier (BTB) (Figure 4B, C).

### 2.5. GC1118 Induced Tumor Cell Apoptosis and Exerted Anti-Angiogenic Effect in Orthotopic PDX Models

Tumor tissues derived from orthotopic xenograft models (N626) were used to investigate whether GC1118 can induce tumor cell apoptosis. Tumor cell apoptosis increased significantly following GC1118 treatment (*p*-value = 0.004, two-sided Wilcoxon rank-sum test; Figure 5); interestingly, the apoptotic effect of GC1118 on GBM tumors was better than that of cetuximab (*p*-value = 0.004, two-sided Wilcoxon rank-sum test).

We also evaluated the anti-angiogenic effect of GC1118 by assessing the microvascular density (MVD) pre- and post-treatment (Figure 5). The vascular endothelial growth factor (VEGF) and EGFR signaling pathways are closely related, and inhibition of VEGF pathways is considered one of mechanism of targeting EGFR [20]. Conversely, overactivation of VEGF pathways independent of EGFR signaling contributes to resistance to anti-EGFR therapy [20]. We observed that the MVD reduced after cetuximab or GC1118, indicating the anti-angiogenic effect of both treatments. The MVD change did not differ significantly between cetuximab and GC1118 treatment (*p*-value = 0.14, two-sided Wilcoxon rank-sum test).

## 3. Discussion

EGFR is a prime target for cancer therapy across various tumor types [9]. EGFR TKIs are approved as first-line therapy for advanced NSCLC with sensitizing mutations [3]. Cetuximab, an anti-EGFR mAb, is the standard of care for metastatic CRC according to the recently updated National Comprehensive Cancer Network (NCCN) guidelines. *EGFR* gene amplification and protein overexpression are characteristic of GBM. However, EGFR-targeting agents have not been approved for GBM treatment yet. GC1118 is a novel anti-EGFR mAb with a high binding affinity to both low- and high-affinity EGFR ligands [21]. Further, GC1118 exerted anti-tumor effects through the suppression of downstream signaling to a greater extent than cetuximab in CRC cell lines [21]. Therefore, we aimed to evaluate the anti-tumor efficacy of GC1118 against GBM using PDX models.

A single-mouse trial scheme was used to mimic the real clinical trial setting. Eight out of fifteen PDX models were sensitive to GC1118. These sensitive PDXs overexpressed EGFR as well as the EGFRvIII variant. While EGFR overexpression did not correlate significantly with GC1118 responsiveness, the EGFRvIII variant was only detected in the sensitive group (Figure 1D), indicating the potential of GC1118 in targeting the majority of GBM tumors with *EGFR* alterations. All these PDXs also responded to cetuximab, which is in agreement with the results of previous studies, demonstrating the anti-tumor effect of cetuximab against GBM harboring the *EGFRvIII* variant [22].

Next, we analyzed the genomic landscape of PDX models to understand the molecular basis underlying GC1118 sensitivity. Various oncogenic pathways contribute to GBM tumorigenesis, and their redundancy has been considered as one of the main reasons responsible for resistance to anti-EGFR therapy [17]. However, we could not identify any specific genomic alteration associated with GC1118 sensitivity (Figure 3). Only *EGFR* gain-of-function alterations, including amplification, may be associated with the anti-tumor effect of GC1118. We conducted high-throughput drug screening using PDCs with available *EGFR* copy number data and observed that *EGFR* amplification was associated with the anti-tumor effect of GC1118 (Figure 3C). This result is consistent with previous studies on cetuximab in preclinical settings [23,24]. This observation is reasonable considering the mechanism of action of mAbs targeting the receptors. The role of *EGFR* amplification status as a biomarker for anti-EGFR treatment was not established in several clinical trials [11,13]. However, recent data from a phase II clinical trial of ABT-414 for recurrent GBM are promising and support the use of *EGFR* amplification as a potential biomarker for predicting sensitivity to anti-EGFR therapy [25].

One of the major issues for treating GBM is the delivery of drugs by overcoming the BBB and BTB. The large molecular weight of antibodies may hinder their ability to penetrate the BBB and/or BTB, consequently limiting treatment efficacy against brain malignancies. Tumor uptake of GC1118 observed in this study indicates that this mAb can cross the BBB of GBM tumors and/or the BBB is sufficiently disrupted to enable its uptake. Accumulation of GC1118 during the first 48 h after drug administration indicates negligible GC1118 efflux. Improvement in the survival of GC1118-treated intracranial models further corroborates the ability of GC1118 to cross the BBB. Whether cetuximab can penetrate the BBB has not been extensively investigated. However, the inferior survival outcomes of intracranial xenografts than dose of subcutaneous models may be indicative of its limited ability to cross the BBB [23].

In the present study, we used cetuximab as a positive control for GC1118. We observed a weak correlation between the TIRs of these two antibodies despite their shared mechanism of action (rho = 0.57, *p*-value = 0.03, Pearson correlation test, Appendix A). Furthermore, we identified that some PDXs were more sensitive to GC1118 (G698 and G500), while other PDXs were more sensitive to cetuximab (G316, and G930). These results suggested a distinct therapeutic vulnerability of GBM tumors to these mAbs.

As mentioned earlier, GC1118 binds to a distinct epitope, and thus GC1118 can be a therapeutic option for *KRAS*-mutant CRCs, which express high-affinity ligands [14]. We compared the mRNA expression of EGFR ligands and observed that all GBM tumors were enriched with high-affinity ligands (Appendix A). BTC and TGFα levels were slightly higher in the GC1118-only sensitive group than in the other group, albeit this was not statistically significant.

Using transcriptomic data, we investigated differentially expressed genes between the GC1118-only vs. the cetuximab-only sensitive group. Interestingly, several *NKG2* family genes were highly expressed in the GC1118-only sensitive group (Appendix A). *KLRC2*, the gene most upregulated in the GC1118-only sensitive group, encodes NKG2C expressed in natural killer (NK) cells, which is involved in NK cell activation [26]. Antibody-dependent cellular cytotoxicity (ADCC) is a well-characterized mechanism of mAb-mediated anti-tumor immunity and usually involves NK cells as the main effectors [27]. Only one study illustrated the ADCC activity of cetuximab in GBM tumors and suggested that the binding affinity of mAb to the extracellular domain of EGFR might determine ADCC activity [22]. Although speculative, differences in ADCC activity between distinct anti-EGFR mAbs might underlie the differences in therapeutic vulnerabilities of GBM tumors. Considering the recent attention paid to immunotherapy, the immunological function of anti-EGFR mAbs requires further investigation, and future studies focusing on the ADCC activity of GC1118 may unravel the immunological function of this mAb.

## 4. Materials and Methods

### 4.1. Glioblastoma Specimens and Their Derivative Cells

After receiving informed consent, glioblastoma specimens were obtained from patient undergoing surgical resection at Samsung Medical Center in accordance with its institutional review board (IRB file no. 2010-04-004). Pathologic diagnosis of GBM has been confirmed by an experienced neuropathologist based on the WHO 2016 criteria. Surgical samples measuring approximately 5 × 5 × 5 mm^3^ were snap-frozen using liquid nitrogen for genomic analysis. Portions of the surgical samples were enzymatically dissociated into single cells, following the procedures previously described [28]. Tumor cells were cultured in neurobasal medium with N2 and B27 supplements (0.5×, each), human recombinant basic fibroblast growth factor (bFG), and epidermal growth factor (EGF). The PDCs used here had shown no obvious contamination of mycoplasma.

### 4.2. Proliferation Assay

Proliferation assays were conducted with EZ-cytox cell viability assay kit (Daeil Lab Service, Seoul, Korea), according to the manufacturer’s instruction. To perform cell proliferation assays, cells were counted and plated with 1 × 10^4^ cell numbers for GBM PDCs in 96-well plates. A total of 4 h after cell plating, GC1118 and cetuximab in cell culture media were treated with 100 µM. After 6 days, EZ-Cytox reagent was added to each well and incubated for 2 h. Light absorbance at wavelength 450 nm was measured using a spectrophotometer. Results from five experiments were analyzed.

### 4.3. Single-Mouse Trial

All in vivo experiments were conducted in accordance with the guideline of the Association for Assessment and Accreditation of Laboratory Animal Care, from the Samsung Medical Center Animal Use and Care Committee (IRB file No. 20160307001). We propagated GBM PDXs to evaluate the therapeutic efficacy of GC1118 and cetuximab by implanting patient-derived tumor cells into the flanks of 6–8-week-old female BALB/c nude mice, which were purchased from Orient Bio Inc. (Seongnam, Korea). Treatment initiated when tumor size was approximately 150–200 mm^3^. PDXs stratified by tumor volumes were randomly assigned to each treatment group to minimize the sampling bias; single PDX was assigned to each treatment group, respectively. The start of dosing was defined as day 1, and tumor volume and body weight were measured twice a week. Tumor volume (mm^3^) was calculated as (length × width^2^) × 0.5. 50 mg/kg of GC1118 or cetuximab (~1 mg/mouse) was injected to PDXs intraperitoneally twice a week (every Monday and Thursday), and equivalent volume of PBS was administered as a control within the same schedule. Mice were monitored daily for any signs of toxicity.

### 4.4. Orthotopic Xenograft PDX Model

To establish orthotopic xenograft models, 6–8-week-old female BALB/c-nude mice (Orient Bio Inc., Seongnam, Korea) were used for intracranial implantation. Patient-derived GBM tumor cells were dissociated and suspended with Hank’s Balanced Salt solution (HBSS; 14170-122, Gibco). A total of 2 × 10^5^ cells (5 µL) were injected intracranially into each mouse using a rodent stereotactic frame (coordinates: anterior/posterior +0.5 mm; medial/lateral +1.7 mm; dorsal/ventral −3.2 mm). Orthotopic PDXs were stratified by body weight and then randomly assigned to distinct treatment groups; 10 mice per each treatment group (GC1118, cetuximab, and control), respectively. All treatments were administered until euthanasia. Animals showing moribund status or loss of body weight >20% were sacrificed. Whole brains were extracted and formalin-fixed to make a paraffin-embedded tissue (FFPE) block for histologic analysis. Survival analysis and IHC studies were performed under blinded inspection.

### 4.5. High-Throughput Drug Screening Experiment

PDCs grown in serum-free medium were seeded in 384-well plates at a density of 500 cells per well in duplicate or triplicate for each treatment. Two hours after plating, PDCs were treated with drugs in a four-fold and seven-point serial dilution series from 20 µM to 4.88 nM using the Janus Automated Workstation (PerkinElmer, Waltham, MA, United States). After 6 days of incubation at 37 °C in a 5% CO_2_ humidified incubator, cell viability was analyzed using an ATP-monitoring system based on firefly luciferase (ATPLit 1step, PerkinElmer). PBS was also included as a control in each plate. Controls were used for the calculation of relative cell viability for each plate, and normalization was performed on a per-plate basis. Dose–response curve (DRC) fitting was performed using GraphPad Prism 5 (GraphPad Software, San Diego, CA, USA) and evaluated by measuring the AUC of the DRC. After normalization, best-fit lines were determined and the AUC value of each curve was calculated using GraphPad prism, ignoring regions defined by fewer than two peaks.

### 4.6. Western Blot Analysis

All tissues were lysed in NP40 buffer (50 mM Tris, pH 7.4, 250 mM NaCl, 5 mM EDTA, 50 mM NaF, 1 mM Na3VO4, 1% Nonidet P-40 and 0.02% NaN3) supplemented with protease inhibitor cocktail tablets (Sigma-Aldrich, St. Louis, MO, USA) and phenylmethanesulfonyl fluoride (PMSF) (Sigma-Aldrich, St. Louis, MO, USA). Equal amounts of proteins were subjected to sodium dodecyl-sulfate polyacrylamide gel electrophoresis (SDS-PAGE) and transferred to polyvinylidene fluoride membranes (Millipore, Burlington, MA, United States). After blocking non-specific binding with 5% skimmed milk or 5% bovine serum albumin (BSA) (Sigma-Aldrich) for 2 h at room temperature, the membranes were incubated with the indicated primary antibodies overnight at 4 °C, followed by incubation with the appropriate secondary antibodies for 1 h at room temperature. EGFR expression was confirmed using rabbit monoclonal antibodies (#2232, Cell Signaling Technology, Danvers, MA, USA) and anti-β-actin (Abcam, Cambridge, MA, USA) antibody. The Amersham ECL Prime Western blotting detection reagent (GE Healthcare, Anaheim, CA, USA) was used for chemiluminescent detection. Image J software (NIH, Bethesda, MD, USA) was used for quantification of the signal intensity of each protein band. The signal intensities of protein bands were normalized to that of β-actin.

### 4.7. In Vivo mAb Distribution Analysis

Orthotopic models were sacrificed at different time points following GC1118 administration. FFPE brain tissue was cut into 4-μm thick sections. Slices of the sectioned brain tissue were deparaffinized, rehydrated, and stained H&E (Sigma-Aldrich). For IHC, the sectioned slides were stained using a goat anti-human secondary antibody (1:200, BA-3000, Vector, Burlingame, CA, USA). Signal was developed with 3,3′-diaminobenzidine (DAB) (Sigma-Aldrich) Sections were counterstained with Mayer’s hematoxylin, dehydrated, and mounted following the manufacturer’s protocol. For quantitative analysis of IHC, images were captured using an automatic histological imaging system. Expression was quantified using HistoQuest analysis software (TissueGnostics GmbH, Vienna, Austria) and the TissueFAXS system (TissueGnostics GmbH, Vienna, Austria) after defining the regions of interest. Several parameters, such as nuclei size and intensity of staining, were adjusted to achieve optimal cell detection. Cells were added to scatter plots based on human-specific marker signals. Cutoff thresholds were determined using the signal intensity of the control. Three different locations were analyzed from the entire slide and the number of positive cells per 1mm^2^ was quantified using the HistoQuest analysis software in the TissueFAXS system. Images were captured under ×200 magnification.

### 4.8. Immunohistochemistry (IHC)

The expression of CD34 (ab8158, abcam) was detected by immunohistochemical staining. In brief, after rehydrated in water, the paraffin sections were placed in citric buffer (pH 6.0) and treated in a microwave. Afterwards, the sections underwent blocking with 5% normal horse serum+ 1% normal goat serum and then primary antibodies were applied (incubated at 4 °C overnight). Then secondary antibodies from HRP-Rat IgG (ab97057, abcam) were applied. Signal was developed with DAB (Sigma-Aldrich). All procedures were performed in accordance with the manufacturer’s recommendations. Apoptosis was determined on paraffin-embedded sections (4-mm thick) by the TUNEL assay method with the use of an ApopTag Peroxidase In Situ Apoptosis Detection kit (S7100, Millipore, CA, USA), according to the manufacturer’s instruction. For quantification, the number of TUNEL-positive nuclei per field was counted manually. Three representative fields per tissue sample from each mouse were scored, yielding a ratio apoptosis value. Graphs represent the mean values of ratio apoptosis (mean ± standard error) compared to control. The IHC images of the tumor sections were obtained using the Aperio imaging system (Leica Biosystems, Wetzlar, Germany).

### 4.9. Next Generation Sequencing (NGS)—Somatic Mutation

Genomic DNA was extracted from fresh frozen tissue specimens using the QIAamp DNA kit (Qiagen, Valencia, CA, USA). The sequenced reads from the FASTQ files were mapped on the human genome assembly (hg19) and aligned using Burrows–Wheeler Aligner [29]. BAM files were preprocessed for sorting, removing of duplicate reads, realigning reads around potential small indels using Picard, GATK and dbSNP, respectively. Furthermore, we used SAMtools to generate and evaluate the realignment and recalibrating base quality score. After the conventional preprocessing of the initial aligned BAM file, we generated mutation—MuTect [30] and SomaticIndelDetector [31] were used to predict the confidence for somatic mutations from tumor and normal tissue pairs. A Variant Effect Predictor (VEP) was used to annotate somatic mutations with potential functional implications and other important information [32].

### 4.10. Target Sequencing

GliomaSCAN is Glioma-specific targeted NGS panel that is capable of capturing single nucleotide variations (SNVs) and insertions/deletions, copy number variations (CNVs), and selected promoter mutations and structural variations that cover a subset of intron regions in 232 essential glioma-related genes. Clinical concordance of GliomaScan was validated using multi-modal methods described previously [33].

### 4.11. Copy Number Variation

For copy number analysis, we used the ngCGH python package version 0.4.4 to generate array comparative genomic hybridization (aCGH)-like data using whole exome sequencing data which were used as the reference for estimating fold changes in copy number in tumors. Tumors and matched normal blood were used to generate gene-based read counts. Normalized copy number values were calculated using a log2 scale.

### 4.12. Whole Transcriptome Sequencing and Analysis

RNA sequencing libraries were generated using the Illumina TruSeq RNA sample Library Preparation Kit. RNA-seq data were used to evaluate mRNA expression levels. Sequenced reads of FASTQ files were mapped to hg19 using GSNAP. Normalized gene expression was calculated and quantified in the form of Reads Per Kilobase of transcripts per Million mappings (RPKM). For further processing, we used the R package DEGseq and RefSeq gene annotation {O’Leary, 2016 #42}.

### 4.13. Statistical Analysis

All statistical analyses were performed using R version 3.6.3 [34]. Continuous variables were compared using the independent sample *t*-test or the two-sided Wilcox test, and categorical variables were tested using the chi-square test or Fisher’s exact test. When comparing the continuous variables with multiple categories (more than three categories), an ANOVA or the Kruskal–Wallis test was used where appropriate. Survival analysis was performed using a Kaplan–Meier plot, and the Log-rank test was used to show statistical differences between survival curves.

Continuous variables are reported as mean ± standard error (s.e.). *p* ≤ 0.05 was used as a threshold for significance. P-values derived from multiple comparisons were appropriately corrected using the false discovery rate (FDR) method. *p*-Values were presented in the following format: non-significant, *p* > 0.05; *, *p* ≤ 0.05; **, *p* ≤ 0.01; ***, *p* ≤ 0.001.

## 5. Conclusions

In the present study, we validated the anti-tumor efficacy of GC1118 in preclinical PDX models. GC1118 exhibited an anti-tumor effect in GBM tumors expressing the EGFRvIII variant. A possible correlation between EGFR amplification and the anti-tumor efficacy of GC1118 was observed, suggesting that EGFR amplification may be a potential biomarker for guiding treatment choices. Furthermore, we observed a survival benefit in GC1118-treated intracranial xenograft models and confirmed the presence of GC1118 within the brain and even in the tumor core. These results support the clinical potential of GC1118 in treating patients with GBM. Based on these findings, a phase II trial of GC1118 for recurrent GBM patients with EGFR amplification is underway (NCT03618667).

## Figures and Tables

**Figure 1 cancers-12-03210-f001:**
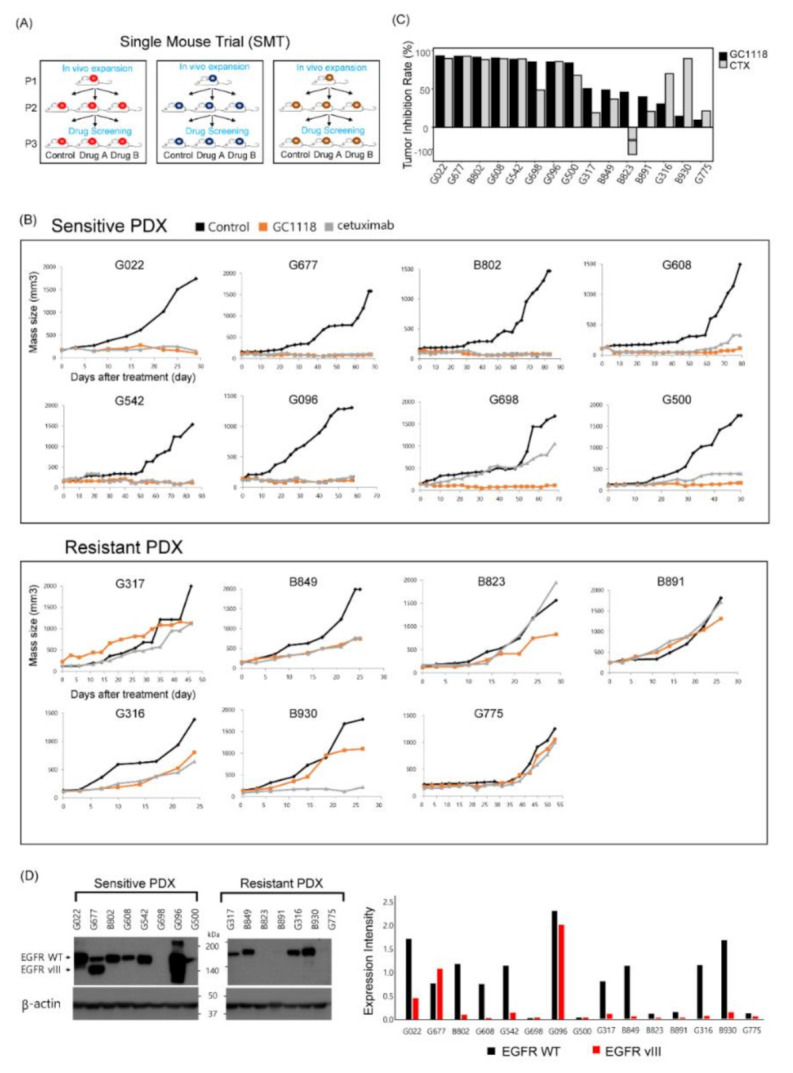
Single-mouse trial to evaluate the anti-tumor efficacy of GC1118 against glioblastoma (GBM) using patient-derived xenograft (PDX) models. (**A**) A schematic illustration of the study design to evaluate the anti-tumor effect of GC1118 using GBM PDXs; (**B**) the anti-tumor effect of GC1118 in xenograft models. Mice were treated with phosphate-buffered saline alone, cetuximab (50 mg/kg), or GC1118 (50 mg/kg) twice a week. Tumor volume was measured twice a week. GC1118 sensitivity was determined according to the tumor inhibition rate as defined below; (**C**) tumor inhibition rate as a measure of anti-tumor efficacy (tumor inhibition rate (TIR): ((1-(volume of treated tumor)/(volume of control tumor)) × 100(%)). PDX models with TIR exceeding 50% were defined as sensitive to treatment; (**D**) expression levels of wild-type (WT) epidermal growth factor receptor (EGFR) and EGFRvIII in PDXs detected using Western blotting. Bar plot represents expression levels of WT EGFR and EGFRvIII normalized to that of β-actin.

**Figure 2 cancers-12-03210-f002:**
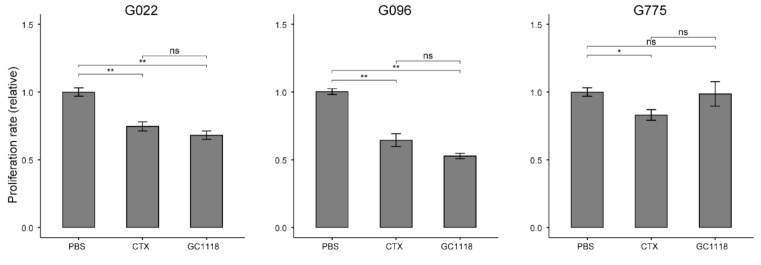
In vitro proliferation assay of glioblastoma patient-derived cells (PDCs). Three different PDCs were used to evaluate the anti-proliferative effect of GC1118. Data are representative of five independent experiments and values are expressed in mean ± standard error p-values were calculated using two-sided Wilcoxon rank sum test (non-significant., *p* > 0.05; *, *p* ≤ 0.05; **, *p* ≤ 0.01).

**Figure 3 cancers-12-03210-f003:**
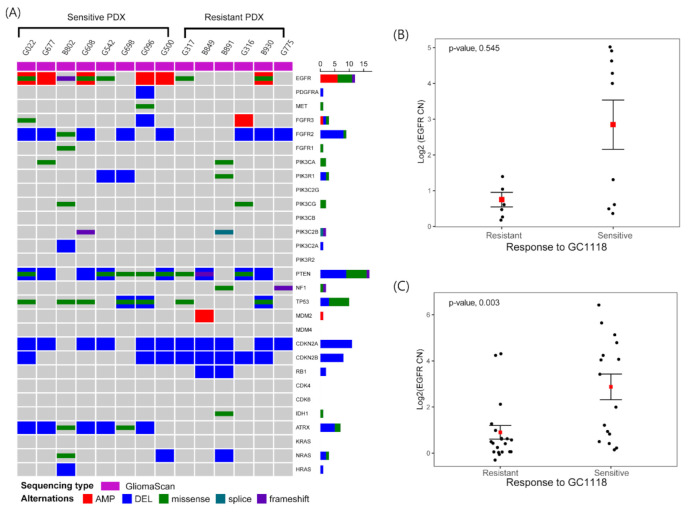
*EGFR* amplification as a potential biomarker to predict the efficacy of GC1118 against GBM. (**A**) An oncoplot depicting major genomic alterations associated with the response to EGFR-targeting therapy; (**B**) comparison of *EGFR* copy number between GC1118-sensitive and -resistant PDX models (genomic data were derived from PDX models); (**C**) comparison of *EGFR* copy number between GC1118-sensitive and -resistant PDCs validated using high-throughput drug screening experiment (genomic data were derived from patient-derived tumor tissue). Mean (red point) and standard error (line range) values are shown.

**Figure 4 cancers-12-03210-f004:**
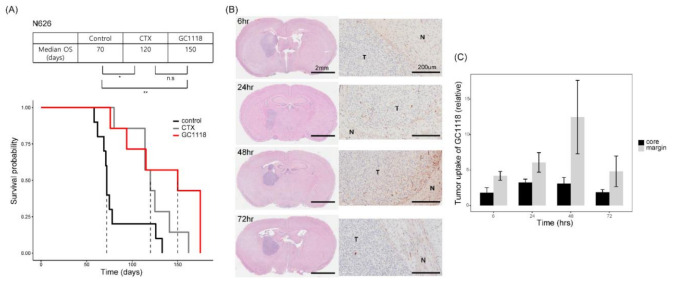
GC1118 can cross the blood–brain barrier and reach the GBM tumor core. (**A**) Kaplan–Meier survival plots of intracranial xenograft models treated with GC1118, cetuximab, or PBS alone (two-sided Log-rank test; non-significant, *p* > 0.05; *, *p* ≤ 0.05; **, *p* ≤ 0.01); (**B**) in vivo distribution analysis of GC1118 within the brain. Intracranial xenografts were sacrificed at several time points following intraperitoneal injection of GC1118. GC1118 at tumor core and margin appeared brown color after 3,3′-diaminobenzidine (DAB) staining (T, tumor; N, normal). Representative H&E staining (scale bar, 2 mm) and immunohistochemistry (IHC) images (scale bar, 200 µm); (**C**) tumor uptake of GC1118 at tumor core and margin in IHC was quantified using Tissue FAXS system. Data are shown as the mean values of three experimental replicates ± standard error

**Figure 5 cancers-12-03210-f005:**
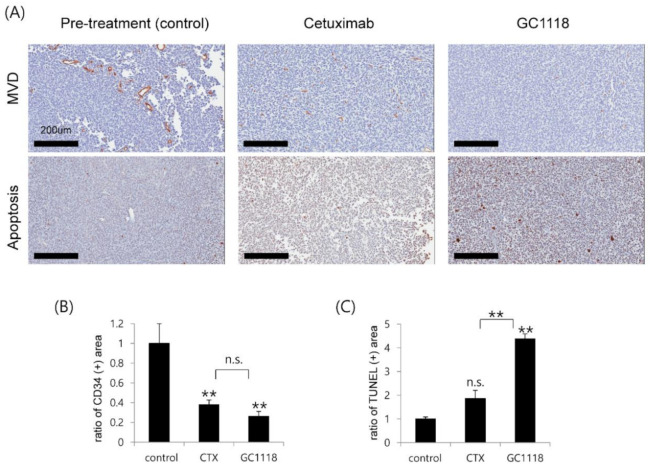
GC1118 exhibited anti-angiogenic effect on GBM tumors and induced apoptosis of GBM tumor cells. (**A**) Representative IHC images of tumor tissue derived from orthotopic PDX models (N626) demonstrated the anti-angiogenic and apoptotic effect of GC1118 treatment. Microvascular density (top row) and apoptosis (bottom row) were assessed using IHC (CD34 for microvascular density (MVD)); terminal deoxynucleotidyl transferase dUTP nick end labeling (TUNEL) for apoptosis on pre- and post-treatment (cetuximab and GC1118) tumor specimen. (**B**, **C**) Data are shown as the means of at least six experimental replicates ± standard error. *p*-values were calculated using two-sided Wilcoxon test (non-significant., *p* > 0.05; **, *p* ≤ 0.01).

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
