# Peer review of "Therapeutic Efficacy of GC1118, a Novel Anti-EGFR Antibody, against Glioblastoma with High EGFR Amplification in Patient-Derived Xenografts"

_cancers, 2020, doi:10.3390/cancers12113210_

Round 1

Reviewer 1 Report

Very good, I have no further suggestions for improvement.

Author Response

We appreciate the positive evaluation of our work by reviewer 1.

Reviewer 2 Report

Lee and coworkers have described the anti-tumor effect of the GC1118 mAb targeting EGFR in glioblastoma patient-derived xenograft. The paper is well-structured and presents thoughtful experiments using numerous patient-derived xenograft. The authors focus on a scientifically important field. Glioblastoma is one of the most deadly cancers, but current therapies with EGFR inhibitors fail. The results presented in the current manuscript may shed new light on potential new treatment option in glioblastoma using a new EGFR inhibitor.

All major points raised by the reviewer are addressed. However, the experimental set up of the new MTT data is unclear. 

In the figure legends of Fig.2 it is stated that "Data are shown as the mean values of at least five experimental replicates", indicating that the experiment is performed once with 5 technical quintuplicates.

In line 320 it is stated that "Results from 3 times experiments were analyzed." 

Straight-forward in vitro experiments should be performed in 3 independent experiments, presenting the mean±st.dev./SE. 

Author Response

We apology for this confusion. Five independent experiments were conducted for each PDC. We revised the figure legend and Method to clarify the meaning.

#figure legends:

In vitro proliferation assay of glioblastoma patient-derived cells (PDCs). Three different PDCs were used to evaluate the anti-proliferative effect of GC1118. Data are shown as the mean values of at least five experimental replicates ± s.e. p-values were calculated using two-sided Wilcoxon rank sum test (n.s., p > 0.05; *, p ≤ 0.05; **, p ≤ 0.01; and ***, p ≤ 0.001).

-->

In vitro proliferation assay of glioblastoma patient-derived cells (PDCs). Three different PDCs were used to evaluate the anti-proliferative effect of GC1118. Data are representative of five independent experiments and values are expressed in mean± s.e. p-values were calculated using two-sided Wilcoxon rank sum test (n.s., p > 0.05; *, p ≤ 0.05; **, p ≤ 0.01; and ***, p ≤ 0.001).

#methods:

Proliferation assay

Proliferation assays were conducted with EZ-cytox cell viability assay kit (Daeil Lab Service, Seoul, Korea), according to the manufacturer’s instruction. To perform cell proliferation assays, cells were counted and plated with 1x104 cell numbers for GBM PDCs in 96-well plates. 4 hours after cell plating, GC1118 and cetuximab in cell culture media were treated with 100 uM. After 6 days, EZ-Cytox reagent was added to each well and incubated for 2 hours. Light absorbance at wavelength 450 nm was measured using a spectrophotometer. Results from 3 times experiments were analyzed.

-->

Proliferation assay

Proliferation assays were conducted with EZ-cytox cell viability assay kit (Daeil Lab Service, Seoul, Korea), according to the manufacturer’s instruction. To perform cell proliferation assays, cells were counted and plated with 1x104 cell numbers for GBM PDCs in 96-well plates. 4 hours after cell plating, GC1118 and cetuximab in cell culture media were treated with 100 uM. After 6 days, EZ-Cytox reagent was added to each well and incubated for 2 hours. Light absorbance at wavelength 450 nm was measured using a spectrophotometer. Results from five times experiments were analyzed.

This manuscript is a resubmission of an earlier submission. The following is a list of the peer review reports and author responses from that submission.

Round 1

Reviewer 1 Report

Therapeutic efficacy of GC1118, a novel anti-EGFR antibody, against glioblastoma with high EGFR amplification in patient-derived xenografts_Cancers_922860

The manuscript from Lee et al. dissects the anti-tumor effect of GC1118 antibody targeting EGFR in glioblastoma xenografts.

The authors focus on a scientifically important field. Structure of the paper is logic and well organized. Due to the high mortality rate of solid malignancies in adults the manuscript fits in the scope of Cancers journal.

I have the following major comments:

  1. The authors should demonstrate the effect of GC1118 on the growth rate GBM cells by a viability assay like MTS/MTT/Cell titer Glo. A PI/AnnV flow cytometric assay would be also informative. Alternatively, a clonogenic assay should be performed. Has GC1118 a direct apoptotic affect on GBM cells or just slows down the cell division? A simple cell cycle flow cytometric assay should be performed. Due to the limitation of the PDX models, cell lines (1 cell line is not enough for Cancers journal) with sensitivity to GC1118 should be enough for these experiments.
  2. Nude mice lack T-cells but NK-cell mediated ADCC could occur. Even, the antibody dependent complement mediated cell death can be the cause of slower tumor growth. The authors should perform an experiment clarifying the role of immune mediated tumor cell death. Injecting 3 sensitive PDX into NSG mice could exclude the NK mediated cell death if the tumors grow with slower rate compared to untreated.
  3. The authors performed IHC detecting the GC1118 antibody. The authors should stain FFPE sections for CD34 (or any relevant marker) to demonstrate the rate of angiogenesis.
  4. The p value is 0.543 in Fig2B but in the relevant text it is 0.3 (line 129). Please check it.
  5. Number of what in Fig 1A from 0 to 15? Copy number of EGFR mutations?
  6. Please improve the resolution of Fig 3B and draw the scale bar.

Minor comments

  1. I suggest the citation and short summary of the previous work with GC1118 in the Introduction.
  2. Check the equation in the line 86, please and reformat it. The equation in the legend of Figure 1 in the line 102 is OK.
  3. Write overexpression in the line 89 because basal expression of wt EGFR is physiologic in the normal tissues.
  4. The legend in Fig1b is not visible, use bigger font size.
  5. Write Expression intensity on the Y-axis in Figure 1D.
  6. Line 112 redundant is Redundant, ….insert ”the”: in GBMs are one of “the” significant…
  7. Write “relevant” instead of representative in line 113
  8. Line 122 …….alterations with”_”gain of function “mutation”…….

Reviewer 2 Report

Very good and careful work.

1. After reading it, you are convinced that it is worth testing this therapy further.

The study shows that the new antibody works at least as well as previous successful therapeutic approaches, especially cetuximab. Whether it is better has to be shown in further investigations as announced by the authors.

I propose to mention exactly this in the abstract with a single sentence.

2. Are there any observations of side effects in the test animals? If so these should be mentioned briefly, possibly in comparison with cetuximab.

Reviewer 3 Report

The authors present preclinical information regarding potential efficacy of an anti-EGFR monoclonal antibody with a distinct epitope, binding to both low affinity and-high affinity EGFR ligands. Testing was performed with subcutaneous patient derived xenografts implanted in nude mice. Treatment effect was detected in some but not all implants. There were no specific predictors of sensitivity detected. Establishment of an intracranial xenograft model resulted in detection of the antibody within the brain.

Methods: no comments; standard xenograft and therapeutic models applied. No comments with regard to intracranial models and detection of antibodies.

Results: no specific comments

Figures and tables: no major issues. The key to figure 1(B) is difficult to find and would recommend relocation.

Discussion: straightforward. I would like to see a better explanation of why the authors believe that the antibody penetrates the blood brain barrier instead of taking advantage of an already disrupted blood brain barrier.